# Systems Biology and Multi-Omics Integration: Viewpoints from the Metabolomics Research Community

**DOI:** 10.3390/metabo9040076

**Published:** 2019-04-18

**Authors:** Farhana R. Pinu, David J. Beale, Amy M. Paten, Konstantinos Kouremenos, Sanjay Swarup, Horst J. Schirra, David Wishart

**Affiliations:** 1The New Zealand Institute for Plant and Food Research Limited, Private Bag 92169, Auckland 1142, New Zealand; 2Land and Water, Commonwealth Scientific and Industrial Research Organization (CSIRO), Ecosciences Precinct, Dutton Park, Dutton Park, QLD 4102, Australia; david.beale@csiro.au; 3Land and Water, Commonwealth Scientific and Industrial Research Organization (CSIRO), Research and Innovation Park, Acton, ACT 2601, Australia; amy.paten@csiro.au; 4Trajan Scientific and Medical, Ringwood, VIC 3134, Australia; kkouremenos@trajanscimed.com; 5Bio21 Institute, The University of Melbourne, Parkville, VIC 3010, Australia; 6Department of Biological Sciences, National University of Singapore, Singapore 117411, Singapore; sanjay@nus.edu.sg; 7Centre for Advanced Imaging, The University of Queensland, St Lucia, QLD 4072, Australia; h.schirra@uq.edu.au; 8Department of Biological Sciences, University of Alberta, Edmonton, AB T6G 2E8, Canada; dwishart@ualberta.ca; 9Department of Computing Science, University of Alberta, Edmonton, AB T6G 2E8, Canada

**Keywords:** mathematical modeling, data analysis, data integration, experimental design, quantitative omics, databases, translational metabolomics, pathway analysis, metabolic networks

## Abstract

The use of multiple omics techniques (i.e., genomics, transcriptomics, proteomics, and metabolomics) is becoming increasingly popular in all facets of life science. Omics techniques provide a more holistic molecular perspective of studied biological systems compared to traditional approaches. However, due to their inherent data differences, integrating multiple omics platforms remains an ongoing challenge for many researchers. As metabolites represent the downstream products of multiple interactions between genes, transcripts, and proteins, metabolomics, the tools and approaches routinely used in this field could assist with the integration of these complex multi-omics data sets. The question is, how? Here we provide some answers (in terms of methods, software tools and databases) along with a variety of recommendations and a list of continuing challenges as identified during a peer session on multi-omics integration that was held at the recent ‘Australian and New Zealand Metabolomics Conference’ (ANZMET 2018) in Auckland, New Zealand (Sept. 2018). We envisage that this document will serve as a guide to metabolomics researchers and other members of the community wishing to perform multi-omics studies. We also believe that these ideas may allow the full promise of integrated multi-omics research and, ultimately, of systems biology to be realized.

## 1. Introduction

Systems biology is an interdisciplinary research field that requires the combined contribution of chemists, biologists, mathematicians, physicists, and engineers to untangle the biology of complex living systems by integrating multiple types of quantitative molecular measurements with well-designed mathematical models [1,2]. The premise and promise of systems biology has provided a powerful motivation for scientists to combine the data generated from multiple omics approaches (e.g., genomics, transcriptomics, proteomics, and metabolomics) to create a more holistic understanding of cells, organisms, and communities, relating to their growth, adaptation, development, and progression to disease [3,4,5,6,7,8,9,10,11,12].

Over the past decade technological advancements in next-generation DNA sequencing [13], SNP-chip profiling [14], transcriptome measurements via RNA-seq [15], SWATH-based proteomics [16], and metabolomics via UPLC-MS and GC-MS techniques [17,18] have greatly increased the ease, and significantly reduced the cost, of collecting rich, multi-omics data. As a result, many researchers are now conducting comprehensive multi-omics experiments, and more data scientists are attempting to integrate these data sets to create new and meaningful biological knowledge [19,20,21,22,23,24,25]. In a clinical setting, a growing number of private companies (e.g., www.arrivale.com, www.viome.com, www.molecularyou.com) are now also using the same low-cost, high throughput technologies to offer multi-omics and precision health assessments.

While large-scale omics data are becoming more accessible, and multi-omics studies are becoming much more frequent—real multi-omics integration remains very challenging. This is because many of the specific analytical tools and experimental designs traditionally used for individual omics disciplines (e.g., genomics, transcriptomics, and proteomics) are not sufficiently well-suited to permit proper comparisons or intelligent integration across multiple omics disciplines. For instance, the preferred collection methods, storage techniques, required quantity and choice of biological samples used for genomics studies are often not suited for metabolomics, proteomics or transcriptomics. Similarly, qualitative methods commonly used in transcriptomics and proteomics are not suited to the quantitative methods used in genomics. While transcriptomics and proteomics are increasingly more quantitative (i.e., RNA-seq in transcriptomics and stable labeled isotope tagging in proteomics), it is increasingly pertinent to compare the applicability and accuracy/precision of quantification strategies (e.g., absolute vs. relative quantification). Furthermore, carefully integrated multi-omics data must be ‘deconstructed’ into single data sets before being deposited into omics-specific databases in order to make it publicly available. These issues underline the fact that high-quality multi-omics studies require: 1) proper experimental design, 2) thoughtful selection, preparation, and storage of appropriate biological samples, 3) careful collection of quantitative multi-omics data and associated meta-data, 4) better tools for integration and interpretation of the data, 5) agreed minimum standards for multi-omics methods and meta-data, and 6) new resources for the deposition of intact multi-omics data sets.

Interestingly, many of the experimental, analytical and data integration requirements that are essential for metabolomics studies are actually fully compatible with genomics, transcriptomics and proteomics studies. In other words, due to its closeness to cellular or tissue phenotypes, metabolomics can provide a ‘common denominator’ to the design and analysis of many multi-omics experiments. It also provides broadly useful guidelines for sampling, handling and processing. Therefore, a greater awareness of metabolomics by other omics researchers or by researchers performing multi-omics experiments could significantly improve the quality and utility of integrated-omics research. To take this integration a step further, it will be critical to be able to associate integrated omics data with the meta-data in the context of the studies being undertaken. For instance, data from host-associated microbiomes can also now be integrated with the exometabolome data to better understand the mechanistic bases of their associations [26].

During the most recent Australian and New Zealand Metabolomics Conference (ANZMET 2018) held in Auckland, New Zealand (from 30 August to 1 September, 2018), the listed authors participated in a peer session on ‘systems biology and multi-omics integration’. The peer session was attended by >20 senior researchers and early to mid-career scientists actively working in the field of metabolomics as well as other omics disciplines. This article summarizes many of the viewpoints raised by those who attended the conference and participated in the peer session. It also provides some guidance on how experimental designs, methods and analytical tools used in metabolomics can facilitate multi-omics studies and data integration. 

## 2. Designing Experiments Suitable for Multi-Omics Integration

Massive increases in throughput and spectacular reductions in costs has enabled multi-omics studies to be routinely performed at a scale not previously imagined. Although each omics platform allows a comprehensive survey for a particular molecular phenotype, the cross-talk between multiple molecular layers cannot be properly assessed by a reductionist approach that analyzes each omics layer in isolation [23]. Instead, systems biology approaches that integrate data from different omics levels offer the potential to improve our understanding of their interrelation and combined influence [27]. A conceptual model for designing a systems biology experiment is depicted in Figure 1. The first step for any system biology experiment is to capture prior knowledge and to formulate appropriate, hypothesis-testing questions. This includes reviewing the available literature across all omics platforms and asking specific questions that need to be answered before considering sample size and power calculations for experiments and subsequent analysis. For example, what is the scope of the study? What are the restrictions for the study? What perturbations will be included/controlled? How will the perturbations be measured? What dose(s)/time point(s) are required? What/which omics platforms will provide the most value? What omics platforms are optional noting that not all platforms need to be accessed to constitute a systems biology study nor do they all provide information that is necessarily required? How will experiments be replicated taking into account biological, technical, analytical and environmental replication? Will individuals be analyzed or will biological samples be pooled? What is the scientific rationale for pooling/not pooling? Does the experimental design properly address these questions? If yes, sample size and power calculations can be assessed and experiments can be planned/performed. If no, the design principals need to be re-evaluated until all criteria are fulfilled.

A high quality, well-thought-out experimental design is the key to success for any multi-omics study. As mentioned above, this includes careful consideration of the samples or sample types, the selection or choice of controls, the level of control over external variables, the required quantities (biomass) of the sample(s), the number of biological and technical replicates, and the preparation and storage of the samples. Additionally, other important factors that need to be considered while designing good multi-omics experiments are the meta-information collected about the samples, the selection of omics methods used and the requirements for data storage, bioinformatics, and computing capabilities. Lastly, it is of utmost importance to ensure that sufficient time and financial resources are available to carry out the experiments.

A successful systems biology experiment requires that the multi-omics data should ideally be generated from the same set of samples to allow for direct comparison under the same conditions. However, this is not always possible due to limitations in sample biomass, sample access or financial resources. In some cases, generating multi-omics data from the same set of samples may not be the most appropriate design. For instance, the use of formalin fixed paraffin-embedded (FFPE) tissues is compatible with genomic studies, but is incompatible with transcriptomics and, until recently, proteomic studies. This is because formalin does not halt RNA degradation [28], and it induces protein cross-linking [29]. Furthermore, paraffin interferes with mass spectrometry performance (affecting both proteomics and metabolomics assays). It is only with the recent advancement of MS technology that very deep and quantitative proteomic profiling of FFPE tissue has become achievable. However, until such time that specialized high-end instrumentation is more broadly available, FFPE analysis remains an issue for many researchers looking to perform multi-omics experiments [30,31]. Another example of multi-omics incompatibility can occur over the choice of biological matrix for a given study. For instance, urine may be an ideal bio-fluid for metabolomics studies, but it has a limited number of proteins, RNA and DNA, making urine a poor choice of biological matrix for proteomics, transcriptomics, and genomics studies. However, on the other hand, blood, plasma or tissues are excellent bio-matrices for generating multi-omics data. This is because they can be quickly processed and frozen to prevent rapid degradation of RNA and metabolites, which would render them unusable for transcriptomic or metabolomics studies [32].

Sample collection, processing, and storage requirements (and limitations) need to be factored into any good experimental design. These variables may affect the types of omics analyses that can be undertaken. Some experiments may have logistical limitations that delay or limit immediate freezing, such as field-work or travel-related restrictions. However, FAA (Federal Aviation Administration)-approved commercial solutions are now available for transporting cryo-preserved samples onboard flights. Other limitations may include sample size restrictions (collecting samples that are too large or too small to process at a single time point or by a single individual) or initial sample handling procedures that may influence biomolecule profiles (e.g., handling live animals). Therefore, careful consideration as to how effectively collected samples can be used, particularly for metabolomics and transcriptomic studies, along with the downstream consequences for data analysis and interpretation is required. Other considerations for experimental design and strategies that should also be considered are shown in Table 1. Recognizing and accounting for these effects early in the experimental design stage will help mitigate their impact on results and down-stream data processing. For more information on the relationship between variance in the data, statistical power, the degree of insights one can gain (e.g., the problem of overfitting when developing statistical models), please refer to the following references [33,34,35].

Sample size relative to resource allocation is one of the main factors affecting experimental design in multi-omics studies. That is, how can one ensure that the sample size is adequate to provide the statistical power needed to detect significant differences while still managing scarce resources (time/money)? As a general rule, the number of variables measured or measurable in a given omics experiment often dictates the number of samples that should be collected and measured (Table 1). Nearly all omics experiments measure hundreds to thousands of variables (metabolites, proteins, genes, transcripts and SNPs). Therefore, the number of biological samples should be sufficient to provide the required statistical power. Underpowered studies may not have the ability to detect associations. This can lead to data misinterpretation and generation of many false negatives and false positives. In single omics experiments, larger sample sizes are often required to overcome this problem. Furthermore, different omics technologies often have different sample size requirements. These number of samples required for statistical power also greatly depends on the extent of variation within and between the treatments, which is reflected statistically in the “effect size”, as described below.

As a general rule, fully quantitative studies (targeted metabolomics and proteomics) require fewer samples than non-quantitative studies (untargeted metabolomics and proteomics). This is due to the greater analytical precision/accuracy that is achievable with targeted and fully quantitative omics measurements. Likewise, for omics studies where the effect size is inherently small (e.g., ambiguous disease phenotypes, most SNP or CNV studies) the number of samples often has to be in the tens of thousands, while in other studies where the effect size is inherently large (e.g., strong disease phenotypes or strong omics signals) the number of samples can be much smaller. For example, the number of samples/patients needed to fully identify, understand or model a simple monogenic disorder such a phenylketonuria (PKU) is on the order of one or two [36], while the number of samples/patients needed to fully identify, understand or model a complex disease, such as schizophrenia, may be on the order of hundreds of thousands [27,37]. Power calculations can be a useful tool to help estimate sample numbers. However, the information on effect size needed for this is often not known until a good body of pilot multi-omics data has already been collected.

Based on the multi-omics studies conducted by our laboratories as well as studies described in the literature, we contend that the experimental design requirements of metabolomics experiments are highly compatible with most multi-omics studies. The adoption of metabolomics experimental design requirements by other omics disciplines could greatly facilitate multi-omics research. Metabolomics experiments are highly compatible with multiple bio-sample types including blood, serum, plasma, cells, cell culture and tissues [28]. These same bio-samples are also preferred for transcriptomics, genomics and proteomics studies. For example, samples intended for metabolomics studies must be stored at −80 °C (to prevent enzymatic degradation of the metabolites and also to stop ongoing metabolic activities). This storage protocol for samples also ensures that RNA and protein integrity is maintained for transcriptomics and proteomics studies [28]. The sample processing times for metabolomics must be relatively quick (minutes) due to the rapid metabolism of metabolites in most bio-samples [28]. Rapid sample processing times are also required for transcriptomics and modestly rapid sample processing times are required for proteomics studies. Adhering to metabolomics processing times ensures that all other samples being used in a multi-omics study will not be compromised. Likewise, metabolomics experiments typically require more material than genomic, proteomic or transcriptomic experiments, therefore the collection of sufficient material for a metabolomics experiment will almost guarantee that enough material will be available for all other omics experiments. Metabolites are also particularly sensitive to environmental influences (diurnal cycles, heat, humidity, diet, age, developmental stage and social interactions) and the tracking of sample meta-data is very important to mitigate the effects of environmental confounders. While environmental confounders are generally less significant for transcriptomics and proteomics studies (and almost negligible for genomics studies), the careful tracking of environmental, social, dietary or other meta-data is a wise practice for all multi-omics studies to facilitate transparency and reproducibility.

Multi-omics analyses should ideally be guided by the sampling guidelines listed above and by the experimental hypothesis being tested. For example, metabolomics would be appropriate in an experiment where metabolic changes are expected but may be less useful if the hypothesis is predicting changes in genetic regulation, especially in upstream signal reception and transduction. Phenotypic landscapes reflect the epigenetic, transcriptional, translational and post-translational implementation of an organism’s genomic information. To establish direct causal and functional associations between genotype and phenotype, it is essential to observe intermediate molecular levels, such as transcript, protein and metabolite abundance [21]. Using one type of omics technology to drive an integrated analysis can be a good strategy to effectively link data sets and build evidence for causation rather than just correlating multiple untargeted data sets. One option for multi-omics integration is to use top-down data reduction approaches that employ genomics or transcriptomics data as a basis to predict phenotypic responses and changes in key signaling and metabolic pathways (Figure 2). Targeted metabolomics and/or proteomics analyses can then be used to measure and validate the functional activity of identified pathways. A benefit of this top-down data reduction approach is that the coverage from genomics/transcriptomics data is inherently greater and so changes in regulation, transport and metabolism may be captured more easily. Limitations include cost (sequencing many samples can become expensive) and the fact that the relationship from gene to protein to metabolite is not necessarily proportional. Therefore, expression differences may not always correlate to functional differences.

An alternative approach to top-down data reduction integration is bottom-up data reduction integration approaches, using either global or targeted metabolomics as the starting point to guide other omics analyses (Figure 2). Metabolites represent the endpoint of gene-environment interactions, and so they are closer to, and more representative of, phenotypic differences. In this approach, metabolomics data are used to target subsequent up-stream proteomics or transcriptomics analyses to uncover mechanistic genes or proteins driving the process/es. Metabolite measurements are relatively quick and affordable compared to many other omics measurements, thus making this approach more appealing from a cost-benefit perspective. However, the coverage of the metabolomics layer is often far more limited than the coverage from the genome or transcriptome layer. This modest coverage may limit the systems biology “search space” and thereby limit the interpretation of the final results.

## 3. Multi-omics Data Integration

Data integration has been characterized as the key bottleneck to all multi-omics studies [33]. This is because data integration requires input and interpretation by a diverse range of scientists or specialists, as previously described. Some of these specialists are needed to evaluate the quality and validity of the study (experimental) design, as well as the quality of the data acquired from the instrument. Assuming that the data obtained are of high quality and are properly validated (see Section 2), a number of approaches can be followed for analyzing and interpreting multi-omics data. These include: (1) post-analysis data integration, (2) integrated data analysis, and (3) systems modeling techniques. Post-analysis data integration and integrated data analysis are primarily discovery tools or hypothesis generators that are intended to reveal new insights or provide some high-level mechanistic understanding. Systems modeling techniques are primarily interpretive or hypothesis testing tools intended to describe mechanistic insight mathematically. Systems modeling can also be used to predict system-wide responses or even treatments (e.g., identifying interventions). Here we briefly summarize these three approaches for multi-omics data integration and the methodologies and tools associated with them.

### 3.1. Post-Analysis Data Integration Approaches

In a post-analysis data integration approach, different omics data sets are first analyzed in isolation, and then key features are “networked” or stitched together through the synthesis of significant features at joint nodes in an overall model pathway (Figure 3). This approach has been used in a diverse range of studies, including the assessment of biological wastewater treatment systems [22], the exploration of microbial resistance of marine sediments after an oil spill [39] and studying the permafrost microbial ecosystem [40]. Post-analysis data integration is also used by some precision health or precision nutrition companies to generate client-specific interpretations.

### 3.2. Integrated Data Analysis Approaches

In contrast to post-analysis data integration, the integrated data analysis approach employs specialized tools to merge different omics data sets prior to undertaking any further data analysis and interpretation [41]. This enables the shared similarities between each omics approach and platform to be statistically derived, as opposed to relying on human interpretation or human biases (Figure 3). Table 2 provides a list of tools that are currently available and can be used in integrated omics data analysis. For instance, methods, such as orthogonal two-way projection to latent structures (O2PLS) and its variant O*n*PLS, were developed to extract systematic variation that is common to two (or multiple) sets of omics data [42,43,44,45]. In clinical science, these methods have been used to evaluate combine metabolomics and proteomic data in a xenograft model of human prostate cancer [46], or to interrogate biological interactions between six different omics data sets in asthma [47]. In environmental science, the same techniques have been used to characterize stress responses or adaptations to different day lengths in poplars [48,49] or to investigate marine sediments [50,51]. Other methods for investigating ecosystem homeostasis with multi-omics approaches have been recently reviewed [52]. Furthermore, the web-based platform 3Omics enables the integration of human transcriptomics, proteomics and metabolomics data [36]. In particular, 3Omics generates inter-omics correlation network to aid data visualization. In addition, this “one-click” platform also assists in statistical analyses of the integrated omics data and is able to perform pathway and gene ontology enrichment. Another popular web-based platform for integrated omics data analysis is “MetaboAnalyst” [53]. As shown in Table 2, this tool is able to integrate and analyze metabolomics data with transcriptomics, proteomics, and genomics data and can be used for data generated for a wide range of biological samples including human, animal, plants and microorganisms.

### 3.3. Systems Modeling

In addition to post-analysis data integration and integrated data analysis techniques, the third approach to data integration is also available—namely systems modeling. Systems modeling and simulation techniques are valuable tools for understanding and even predicting the details of complex biological systems [54]. Model-based integration methods rely on a well-defined understanding of the system being investigated in order to compare new experimental findings against modeled predictions. Such an understanding is often based on having comprehensive, pre-existing genomic, transcriptomics and/or metabolomics data on the system being studied. These modeling systems may incorporate dynamic/kinetic models that solve systems of differential, or partial differential equations [55], agent-based kinetic models [56], or Petri-Net models [57] or they may involve steady state models, such as flux-balance models [58]. Interestingly, almost all of these systems modeling approaches are anchored with metabolic reactions and extensive metabolomics data.

Some of the most impressive examples of multi-omics integration and many of the most compelling successes in systems biology have been achieved through systems modeling methods. For example, in the 1990s Palsson and co-workers quantitatively predicted growth and byproduct secretion in *E. coli* by modeling cellular metabolism and developed the concept of flux balance modeling [59,60,61,62]. Subsequently, the first successful attempt to model a living cell kinetically was the E-cell project led by Masanori Tomita in 1999 [63,64]. This project focused on modeling the kinetic dynamics of metabolic pathways and the control of enzyme production through gene expression of *Mycoplasma genitalium*. This single cell model integrated experimental metabolomics data with genomics, transcriptomics and proteomics data. The E-cell concept was later extended to modeling human erythrocyte metabolism [65,66]. Several practical outcomes have been achieved with the E-cell erythrocyte model including an improved understanding of erythrocyte hypoxic responses [66], enhanced blood storage methods [67], and the development of a new nucleo-base solution for extended blood storage [68]. By the mid-2000s particularly impressive simulation work was being done with modeling *E. coli* [69]. This led to the successful prediction of dozens of conditionally essential genes in *E. coli* [70] through metabolic flux-balance modeling. Similar work using Petri-Net models led to the identification of key protein and gene regulators in *E. coli* glycogen metabolism [71]. These activities inspired the development of a variety of open source tools and mark-up languages for system modeling, such as the Systems Biology Markup Language or SBML [72], CellML [73], Cell System Markup Language or CSML [74] as well as modeling tools, such as Cell Illustrator [74].

Extensions from single cells to multi-cellular systems and multi-organ systems started to occur around 2013 with the appearance of Recon 2, a community based global reconstruction of human metabolism [12]. This comprehensive model incorporates cell and tissue-specific metabolomics data, proteomic data and gene expression data for the human body and its many cell types. Through Recon 2 and its later derivatives, it has been possible to model the effects of common drugs on human metabolism [75] to predict the effects of disease-gene mutations [76] and to model disease conditions, such as inflammatory bowel disease [77]. Of particular note is the development of the COBRA (Constraint-Based Reconstruction and Analysis) system for working with Recon 2. COBRA permits integrated modeling of metabolism and macromolecular expression (proteome or transcriptome) at a genome-scale [78,79]. Over the past three years, the COBRA modeling system has been used to create a 3D model of gene variation in human metabolism—Recon3D [80]. COBRA has also been used as a tool to develop models in order to subsequently predict dietary supplements for treating Crohn’s disease [81] and to integrate human and gut microbiome metabolism with nutrition and disease in the Virtual Metabolic Human database [82]. The extension of systems modeling to microbiome studies (where metabolomics is integrated with metagenomics) is becoming particularly popular. For example, Noecker et al. [83] integrated taxonomic and metabolomics data to predict the effects of community ecology on metabolite concentrations. These predictions were evaluated with measured metabolome profiles from the vaginal microbiome and it was concluded that predicted species composition correlated with identified putative metabolic mechanisms [83]. The availability of previously published data on the vaginal microbiome and its metagenomics data were key to the success of this model [84,85,86,87,88].

As seen from the above examples of systems modeling for multi-omics integration, as well as through many other examples, most multi-omics systems models are based on some form of metabolic model or metabolic readout [11,20,83,89,90,91]. This is true even though they invariably include genomics, proteomics and/or transcriptomics data in the model. This fact serves to emphasize the key role that metabolomics must play in multi-omics integration, especially as it relates to systems modeling. The reason why metabolomics plays such an integral role in systems modeling and multi-omics integration is because it can be quantitative. Systems modeling cannot be performed without accurate values or accurate concentrations as inputs and, likewise, systems models cannot be easily verified without accurate, quantitative concentrations as outputs. Metabolomics can deliver both (quantitative input and output data), making it extremely valuable to systems modelers. It should be noted that quantitative proteomics also meets this requirement, albeit at a level that is not as close to the observed phenotype as the metabolome. Therefore, the central challenge for systems modeling is the collection of accurate, quantitative reference data on the genome, the transcriptome, the proteome or the metabolome. As a result, model-based integration is often limited to only those systems that are already well defined (i.e., common model organisms) or systems for which accurate, quantitative multi-omics data have been collected.

### 3.4. Software Tools, Databases and Approaches for Multi-Omics Integration

Biswapriya et al. [92] recently reviewed different tools available for multi-omics data integration in detail. This work highlights the fact that numerous databases, software tools, and approaches are now freely accessible to assist with integrating multi-omics data sets, regardless of the multi-omics data integration approach.

As listed in Table 3, a number of context-specific databases and tools have been developed (i.e., targeted towards the integration of omics data from specific animal models, medical and clinical studies, and selected plant species) and are in use. While most available databases are fairly general, there is a range of databases with a specific focus [93]. For instance, certain species-specific databases are now publicly available that include data on the genome, transcriptome, proteome and/or metabolome of several model organisms. These include the mouse genome database or MGD [94], FlyBase (*Drosophila spp*) [95], WormBase (*Caenorhabditis elegans* and helminth) [96], the *E. coli* metabolome database or ECMDB [97], the EcoCyc database [98], the Yeast metabolome data or YMDB [99], the Plant Metabolic Network or PMN [100] and the *Saccharomyces* genome database or SGD [101], just to name a few. There are also extensive databases on humans that contain rich data on the human genome, proteome, metagenome and/or metabolome. These include the Human Metabolome Database or HMDB [102], Recon 2 [12], Recon3D [80] and the Virtual Metabolic Human database [82]. In addition to these species-specific resources, there are also general multi-species resources on genes and proteins, such as GenBank and UniProt [103,104], multi-species collections on metabolites, such as ChEBI [105], and MetaboLights [106], multi-species collections on lipids, such as Lipid Maps [107], multi-species collections on proteomics or protein expression data, such as PRIDE [108], and multi-species pathway databases, such as KEGG [109], Reactome [110] and MetaCyc [111]. Additional details on these databases are provided in Table 3.

While there are many open access databases to help with multi-omics integration, a variety of open-access tools are also available to help with statistics and visualization of multi-omics data. These include tools to facilitate data quality checks, data normalization and data transformation (e.g., MetaboAnalyst 4.0 and mixOmics). They also include software to assist with performing multivariate statistics, data clustering and data interpretation. Many multi-omics integration tools include methods to generate and view interactive correlation or association maps (hairballs), as well as metabolic and signaling pathways. A number of these tools and databases are listed in Table 2 and Table 3.

## 4. Challenges in Multi-Omics Integration

Meaningful biological interpretation of multi-omics data requires a constant evolution of databases and data analysis tools. Analytical and data analysis platforms have improved substantially over the past decade, however, multiple challenges still exist [139]. Here, we identify and discuss some of the key pitfalls and challenges that appear to be hindering the progress in multi-omics integration and systems biology research. However, we acknowledge that some of these challenges might be solved in the near future while new challenges may arise with time.

### 4.1. The Nature of the Omics Data Sets

Omics data are inherently highly variable and noisy. Furthermore, most omics data are only qualitative in nature, making it very hard to reproduce and even harder to compare. When only qualitative data are available, multi-omics integration, particularly from multiple sources, becomes difficult, if not almost impossible [90,140,141]. While DNA sequence data are often very accurate and highly reproducible (error rates of <0.001%), they are widely considered qualitative with a large number of false positives based on the sheer number of reads obtained: transcriptomic, proteomic, metagenomics and the majority of metabolomics data are highly qualitative and poorly reproducible, as reflected in reported false discovery rates (FDR) and adjusted *p*-values [26,142,143,144,145]. Many of the difficulties that these omics techniques have are inherent to the practices or measurement methods adopted by many platform users. For example, most proteomics researchers do not use well-defined peptide reference standards or isotopically labeled peptide standards for identification or quantification. Likewise, many microarray/transcriptome researchers use inherently qualitative microarray technologies and do not use universal reference materials with precisely known numbers of transcripts. Similarly, most metagenomics labs do not employ standard OTU definitions or standardized 16S-RNA to consistently identify microbes. It is now well-accepted by the microbiome community now that even RNA-Seq counts are not the actual counts, rather they represent relative abundances of those transcripts. Similarly, the vast majority (>80%) of published metabolomics studies are also qualitative, with little or no use of multiple reaction monitoring (MRM) or well-defined metabolite reference standards or isotopically labeled metabolite standards. As a result, omics labs that analyze exactly the same material using similar or identical omics technologies (i.e., comparing one metabolomics/proteomics approach to another comparable metabolomics/proteomics approach) will often get wildly different results [26,142,143,144,145], so much so that in some instances upward trends in one data set, could become downward trends in another.

Even if the problems with the omics technologies could be controlled or managed through improved availability and use of reference standards and standardized operating protocols, there still remains a problem with significant inter-laboratory variations in sample storage, extraction and handling protocols. These issues are often related to the skills, organization, and patience of individual technicians or researchers. Sample-specific factors, such as the population structure in the sample, the cell or tissue composition, inherent sampling biases, batch effects, and other technical artifacts can add further heterogeneity to omics measurements as does the way in which different classes of biomolecules behave and interact (e.g., response time, stability, half-life, regulation).

Precisely measured, quantitative data that is calibrated to standard reference materials, checked against authentic standards, assessed with quality controls and measured with universal standard operating protocols (SOPs) is what is needed for robust, reproducible multi-omics integration. This type of high quality, quantitative data are actually obtainable via genomics, metabolomics, and proteomics. They are potentially obtainable via transcriptomics [146] and meta-genomics [147]. When quantitative data are available, it is possible to perform multi-omics integration. It is also possible to perform robust intra- and inter-lab comparisons (i.e., ring trials). Examples of this can be seen with the impressive results described in the systems modeling section (see Section 3.1) where essentially all of the baseline data sets used for modeling were obtained from open-source quantitative data repositories.

The lack of sufficient meta-data is another key hurdle to the successful integration of multi-omics data sets [89]. Often considerable time, money and effort may go into the collection of molecular omics data, while very little effort goes into collecting the meta-data about the samples, organisms or patients. Meta-data are information about the data, which typically includes where, when and how the data were collected as well as information about the observable phenotypes of the samples. Meta-data are important to enabling reproducibility and biologically relevant interpretation of omics results. For example, in the case of plant multi-omics analysis information about the soil type, soil composition, watering conditions, temperature trends, relative humidity, plant age, season, light levels and plant breed can have a significant impact on metabolite, protein and transcriptome measurements. For cell or tissue culture multi-omics analyses, important meta-data could include information about the cell types, cell sex, cell growth media, cell oxygenation levels, and cell generation. For animal studies, meta-data could include information about age, sex, weight, feeding levels, feed composition, strain or breed, cage location, light conditions, activity, diurnal cycle, health status and proximity to other animals. For human studies, meta-information including age, sex, job type, ethnicity, marital status, BMI, smoking/drinking habits, lifestyle, exercise, diet, health status, drug intake, diurnal cycle, menstrual cycle, and ethnicity should be collected, as they could also affect omics measurements. Meta-data could also be considered in the form of ionomics or metallomics data from the cells or tissues being investigated. Without taking into account these confounders, it is often difficult to identify clear signals in the molecular data or signals may be misinterpreted (e.g., one may misidentify a molecular trend that is simply due to an age difference). If these meta-data are not captured, the collection of large data sets is required to mitigate the sources of variation [148], which may greatly increase the cost of a multi-omics study.

### 4.2. Dispersed Data Sets and Non-Interoperable Tools

There is no shortage of data tools and software available for multi-omics integration, as clearly demonstrated by the size and scope of Table 2. Often, the available multi-omics databases are of very high quality and exceptionally well curated and many of the software tools are well designed and maintained. However, it is clear that researchers (including many of the authors of this manuscript) may be unaware of all of these tools and what they may offer. This problem may be due to the sheer number of resources (now numbering hundreds) or the lack of a central repository which catalogs, links and rates or summarizes these tools. A third reason may be the tendency of researchers to ‘stick with what they know’. This leads to a bias among scientists to treat every problem as a nail that has to be hit with the same hammer. This could be partly caused by the user-unfriendliness of many multi-omics programs. Some are difficult to install, others have limited operating system compatibilities, while others have enormously steep learning curves. Increasing researcher awareness and accessibility to tools (e.g., through central repositories and making tools available over the web or converting them to web-servers) will improve their use and uptake.

Lack of interoperability is another frequently cited problem with bioinformatics tools, not just multi-omics tools. Different bioinformatics and cheminformatics databases have different formats, many of which are non-standard. Similarly, different programs may also require their input data to be in non-standard formats and will output data in a format that is incompatible with other programs. This can make it difficult to create a multi-program or a multi-database workflow and may require users to spend their time writing scripts to convert and reconvert data so that it can be read by other programs. The movement to FAIR (findable, accessible, interoperable, and re-usable) data standards in bioinformatics software and databases could potentially alleviate these challenges [149].

### 4.3. Inadequate Pathway and Data Visualization Tools

While there are many excellent tools for multi-omics integration and visualization (see Table 2), there is clearly still room for improvement. One area where significant work is needed lies in the scope and availability of pathway databases. Pathway databases provide biological context while at the same time providing clear links between genes, proteins, and metabolites. KEGG [109], MetaCyc [111] and Reactome [110] are freely available and widely used pathway databases, however, they limited in portraying what is known about biological pathways or molecular processes. While all three databases emphasize metabolic pathways, they were all developed before the advent of metabolomics (or even proteomics) and thus lack some of the new insights that these technologies bring. It is important to remember that many other kinds of pathways are also important in biology, including: protein signaling, metabolite signaling, gene activation, protein and metabolite transport, disease, drug action and drug or xenobiotic metabolism pathways, as well as many more. Although some of these pathways are captured in commercial resources, such as the Ingenuity Pathway Analysis (IPA) system produced by Qiagen, the high cost, closed nature of the database, and the lack of compatibility with many other bioinformatics software tools makes such commercial tools difficult to integrate into multi-omics pipelines. Certainly, the creation of far more extensive, open access pathway databases with suitable rendering functions and machine-readable features, would make multi-omics integration much easier and far more user-friendly. As in the case of open platforms of open-source R-packages, R-based multi-omics packages can be encouraged through collaborations internationally.

Construction and visualization of network models is another challenging aspect of multi-omics integration. The use of metabolic network models, generated via metabolomics, has greatly facilitated the integration of multi-omics data [150]. These metabolic network models have been extended to genetic regulatory networks, protein-protein interaction network models and metabolic-reaction network models [12,151]. However, these models remain ‘insular’ in that they only allow the modeling a single omics network. Clearly, if network models could be extended to more clearly illustrate or connect multi-omics data, this would greatly enhance visualization and interpretation of multi-omics data. Recently, multi-layer networks—which can be defined as networks formed by multiple omics layers—have been created that allow the rendering of specific interactions between different omics layers [152,153].

### 4.4. Failing to Demonstrate Utility

Multi-omics studies potentially represent the pinnacle of achievement in molecular characterization. However, there are questions around their real value and utility. For example, is a multi-omics signature for diabetes or obesity really useful if the same diagnosis can be made by a simple blood glucose test or having someone step on a weight scale? Likewise, if a single, carefully interpreted, metabolomics study provides just as much information as a comprehensive multi-omics study, is anything useful gained? Currently, many published multi-omics studies appear to simply demonstrate that they are feasible and modestly informative. Some have demonstrated a post-hoc rationale for what was long known or commonly observed while others have provided some new useful mechanistic insights but have not led to new drugs, new therapies, new diagnostics or new biotechnologies. As yet no truly significant or ‘groundbreaking’ results have emerged from multi-omics studies. That is not to say that these types of studies will not have an impact, but for multi-omics research to gain traction with the public and with funding agencies, it will need to demonstrate clear utility and highlight novel insights. Currently, the most tangible and useful applications for multi-omics studies are in the areas of precision medicine, personalized health, or precision nutrition [154,155,156]. These appear modestly useful, and certainly, if larger population studies could be performed, their utility could grow significantly. Useful/practical applications are also appearing in environmental protection and environmental adaptation/resilience, especially after significant and ongoing contamination events or natural disasters [39,50,51,157].

### 4.5. Limited Research Funding

Multi-omics research can be expensive, and access to multiple analytical instrumentations and multidisciplinary specialists requires a considerable level of research funding. Indeed, a robust multi-omics study on a human research population can easily cost in excess of US$500,000. Therefore, the availability of sufficient funding is often the central limiting factor for any multi-omics study. Traditionally, most research funding has been allocated to genomic, metagenomics and transcriptomics studies while proteomics, in most jurisdictions around the world, is only funded at about 10% of the level of genomics and, in most countries, metabolomics funding is usually at about 2% of the funding level of genomics. As a result, most multi-omics studies in humans involve combined genomics and transcriptomics studies or combined genomics and metagenomics/microbiome studies. Multi-omics studies involving metabolomics are relatively rare, yet they are consistently among the most informative or most highly cited [75,76,77]. Clearly, if true multi-omics research is going to be conducted, a re-balancing of research funding is needed. As emphasized throughout this document, we believe that the metabolomics field has a great deal to offer to multi-omics researchers and that metabolomics methods should lie at the core of any truly multi-omics research study.

## 5. Recommendations

Based on the challenges identified above along with some of the potential solutions (achieved by us and others) we have produced a list of recommendations to consider when designing or contemplating a multi-omics study. These are:Adopt the sample collection, preparation and measurement standards used in metabolomics studies. This would ensure high-quality data collection in most multi-omics studies;Measure multi-omics data in a robust, quantitative manner to ensure reproducibility, enforce comparability and permit facile integration;Use reference standards, quality control (QC) samples, and universal standardized operating protocols (SOPs) to enable consistent multi-omics measurements across laboratories;Perform power analyses, where possible, prior to conducting large scale multi-omics studies; and,Collect and record comprehensive meta-data to guide and inform well-designed multi-omics studies (please see [106,158,159,160]).

We have also identified the following as key requirements to support the multi-omics community in its efforts to carry out truly integrated, systems biology research:Create centralized data repositories, curated or reviewed software lists and improved software/database interoperability (adherence to FAIR data standards) to improve multi-omics integration;Improve or develop more comprehensive open source pathway databases and network visualization tools;Increase levels of funding and increase awareness of the need for metabolomics in multi-omics studies;Demonstrate clear utility of multi-omics studies to both the public and funding agencies; and,Undertake more community-driven activities to lead to the creation of multi-omics tools and resources better suited to the community’s needs.

In addition to these above recommendations, we also encourage software developers to take more initiative to develop more user-friendly omics integration of web tools and software. Advancements in machine learning approaches would also be highly beneficial for the integration and interpretation of multi-omics data. Moreover, successful researchers should ensure to pass on their successes or share their stories of multi-omics integration to a much broader audience. Communicating successes or discoveries to the popular press raises awareness and shows both the public and scientific funding agencies that multi-omics research can have a real social impact. The use of social media is changing how we conduct social communication, it could and should change how we conduct our scientific communication too.

## 6. Conclusions

Most of the data generated over the past 25 years by various omics platforms have been qualitative or semi-quantitative in nature. While qualitative data can be used to compare similar variables across samples within a single lab (using a single measurement platform), they do not support comparisons across labs or across platforms. Likewise, qualitative measurements do not allow for consistent and accurate integration of multi-omics data where measurements typically have to be done on multiple platforms, across multiple laboratories, over extended time periods. Careful quantitation of omics data allows much more accurate comparison to reference values, more consistent inter-laboratory comparisons, greater reproducibility, improved interpretability, and far simpler data integration. Indeed, because of the widespread lack of quantification, only one single proteomic assay and only five transcriptomics assays have ever been translated into the clinic [161,162]. On the other hand, quantitative metabolite measurements have enabled the translation of more than 300 chemical tests to the clinic [163], while the use of quantitative genotyping has led to the development of about 75,000 genetic tests [164].

In addition to the importance of generating quantitative omics data, it is also important to make use of publicly available data sets, where omics-specific reporting and higher quality data standards have been followed. Such an approach will benefit the scientific community and reduce the amount of repeated work amongst different groups. The description of a given domain can be represented in any format, but the use of common ontologies makes it easier to compare, integrate and interpret data from similar domains, multiple omics platforms as it facilitates the integration, computational processing and comprehensive interpretation of that data [165].

Lastly, there is a greater need for cross-talk among the different omics communities. This is especially true given the insular or “silo-ed” approach that most omics researchers have traditionally adopted in publishing or presenting their work. There is a tremendous amount that can be learned from shared interactions, particularly from those omics areas that are developing comparatively faster than others. Webinars could be one approach that could inexpensively facilitate such conversations. Likewise, the formation of professional societies whose central aim is integrating multi-omics data could also help break down some of the existing barriers and improve collaboration between the existing professional omics societies and systems biology societies. It could be achieved by, for example, establishing mutual education programs, creating common data standards, forming common task groups and developing cross-omics engagement guidelines that address authorship policies and promote the advancement of early to mid-career researchers. Furthermore, greater connectivity and collaboration between the respective scientific conferences could be encouraged through shared/mutually supported conference sessions.

## Figures and Tables

**Figure 1 metabolites-09-00076-f001:**
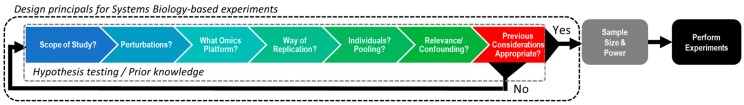
A conceptual model for designing a systems biology experiment.

**Figure 2 metabolites-09-00076-f002:**
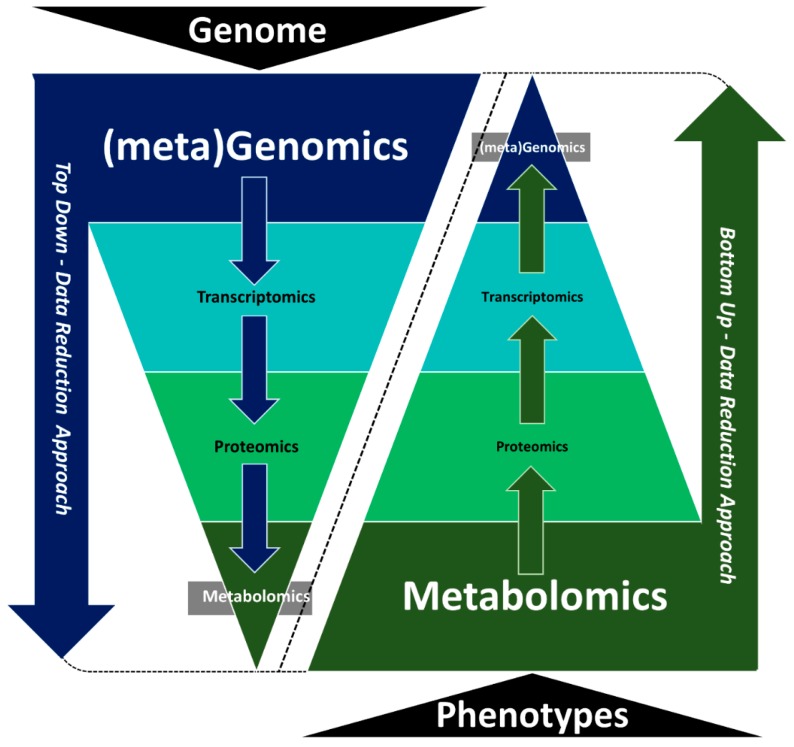
Top-down and bottom-up data reduction integration approaches used in system biology.

**Figure 3 metabolites-09-00076-f003:**
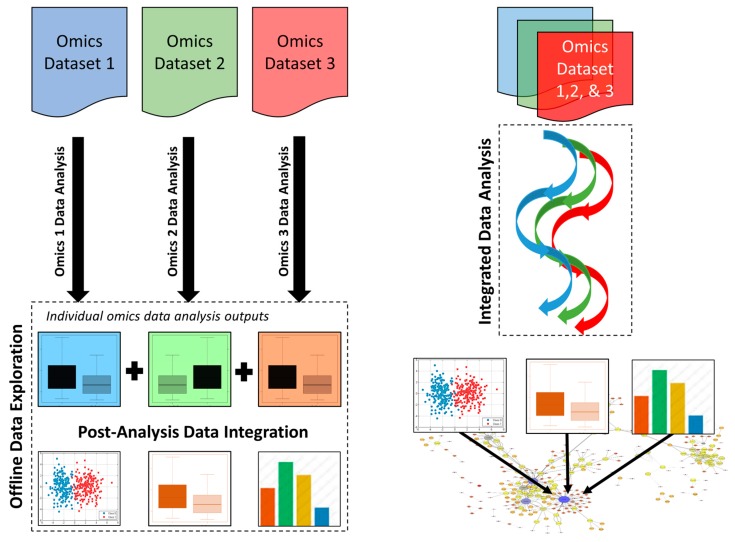
Principal differences between post-analysis data integration (**left**) and integrated data analysis (**right**) for handling multi-omics data sets.

**Table 1 metabolites-09-00076-t001:** Potential limitations while designing multi-omics studies and possible strategies to overcome them.

Potential Limitations	Strategies to Overcome Limitation
Limited biomass of sample small sample or limited accessibility to sample (e.g., single cell, skin or saliva swab) [38]	Pooling Specific methods for small samples (e.g., methods for single cell omics)
Heterogeneity of cell type/ composition (e.g., microbiome community, whole organism, tissue or single cell). Proportions of multiple cell types in a sample can change substantially and shift omics profile [27].	Replication Adequate sample size (n) Homogenization Reference samples/data
Differences in specific biomolecules in sample types (e.g., urine may have many metabolites but very few proteins, DNA and RNA in comparison to blood, stool or tissue samples).	Choose appropriate sample for omics analysis or appropriate omics analyses for sample based on hypothesis.
Technical artifacts, including batch effects.	Reference and quality control (QC) samples Internal standards Randomization Appropriate statistical models (e.g., mixed effects) to account for batch effects [27]
Multiple testing loss of statistical power	Replication Adequate sample size
Background contamination (e.g., in a microbiome study stool samples will have host DNA, RNA, protein, and metabolites)	Specific methods depending on omics analysis and contamination (e.g., rRNA or host depletion for RNA extraction for meta-transcriptomics) Background control/reference samples
Differences in analytical platforms and integrating data sets of multi-omics that measure fundamentally different biomolecules	Standard control samples used across all omics data sets may help to harmonize measurements and variations

**Table 2 metabolites-09-00076-t002:** Summary of multi-omics integration software tools and web applications (adapted from Fondi and Liò [90] and Beale et al. [112]).

Software Tool	Omics Integrated	Domain	Functionality	Type of license	Reference
**3Omics**	Transcriptomics Proteomics Metabolomics	Medical (human)	- Correlation network analysis - Co-expression analysis - Phenotype generation - KEGG/HumanCyc pathway enrichment - GO enrichment - Name to ID conversion	Open	[41]
**BiofOmics**	Transcriptomics Proteomics Metabolomics	Biofilm	- Experiment library - Data depository	Open	[113]
**BioCyc/MetaCyc**	Genomics Proteomics Metabolomics	Unspecified	- Online encyclopedia of metabolism - Predicted metabolic pathways in sequenced genomes - Enzyme data set - Metabolite database	Open	[111]
**Cell Illustrator 5.0**	Genomic Transcriptomics Proteomics	Unspecified	- Draw biological pathway models and simulations - Run biological cellular simulations and graphical display results	Licensed	[74]
**CellML** (Open source XML language)	Transcriptomics Proteomics Metabolomics	Unspecified	- Open source language for biological cellular models	Open	[73]
**COBRA**	Transcriptomics Proteomics Metabolomics Fluxomics	Unspecified	- Genome scale integrated modeling of cell metabolism and macro-molecular expression	Open	[12,79]
**Cytoscape** with **MODAM,** and; **Cytoscape** with **OmicsAnalyzer**	Transcriptomics Proteomics Metabolomics Fluxomics	Unspecified	- Multi-Omic Data Miner and OmicsAnalyzer were designed as an accessible and handy Cytoscape plugin that facilitates omics analysis - Compile all biologically-relevant information regarding the model system through web link associationMap the network components with multi-omics data - Model omics data	Open	[114,115]
**E-Cell**	Transcriptomics Proteomics Metabolomics	Unspecified (Cells)	- Modeling, simulation, and analysis of complex, heterogeneous and multi-scale cellular systems	Open	[63]
**Escher**	Genomics Proteomics Metabolomics	Unspecified	- Web application for visualizing data on biological pathways. - Rapidly design new pathway maps based on user data and genome-scale models - Visualize data related to genes or proteins on the associated reactions and pathways - Identify trends in common genomic data types	Open (MIT license)	[116]
**Gaggle**	Variety of omics platform bioinformatics solutions	Unspecified	- Inoperability of the following tools: - Bioinformatics resource manager - Cytoscape - DataMatrixViewer - KEGG - Genome Browser - MeV - PIPE - BioTapestry - N-Browse	Open	[117]
**GIM3E** (Gene Inactivation Moderated by Metabolism, Metabolomics and Expression)	Transcriptomics Metabolomics	Unspecified	- Establishes metabolite use requirements with metabolomics data - Model-paired transcriptomics data to find experimentally supported solutions - Calculates the turnover (production/consumption) flux of metabolites	Open; Phython based and requires COBRApy 0.2.x.	[118]
**INMEX** (Integrative meta-analysis of expression data)	Transcriptomics Metabolomics	Medical and Clinical	- Meta and integrative analysis of data - Pathway analysis	Open	[119]
**IMPaLA** (Integrated Molecular Pathway Level Analysis)	Transcriptomics Proteomics Metabolomics	Medical and clinical	- Enrichment analysis - Pathway analysis	Academic only	[120]
**Ingenuity Pathway Analysis**	Metagenomics Transcriptomics Proteomics Metabolomics	Medical (human) and clinical.	- Metabolic pathway analysis - Network visualization - Data integration - Upstream regulator analysis - Mechanistic networks - Causal network analysis - Downstream effects analysis	Commercial	[121]
**IOMA** (Integrative Omics-Metabolic Analysis)	Proteomics Metabolomics	Unspecified	- Integrates proteomics and metabolomics data to predict flux distributions	Open	[20]
**KaPPA-View**	Transcriptomics Metabolomics	Plants	- Integrates transcriptomics and metabolomics data to map pathways	Open	[122]
**MADMAX** (Management and analysis database for multiple omics experiments)	Metagenomics Transcriptomics Metabolomics	Plants, Medical and Clinical	- Integrates omics data - Statistical analysis and pathway mapping	Open	[123]
**MapMan**	Metagenomics Transcriptomics Metabolomics	Plants (developed for use with *Arabidopsis*. Includes more species)	- Compare data across these two species - KEGG classification - Classification into KOG clusters - Mapping expression responses	Open	[124,125]
**MarVis-Pathway** (Marker Visualization Pathway)	Metagenomics Transcriptomics Metabolomics	Unspecified	- Toolbox for interactive ranking, filtering, combination, clustering, visualization, and functional analysis of data sets containing intensity-based profile vectors	Academic only	[126]
**MassTrix**	Transcriptomics Metabolomics	Unspecified	- Integration of data - Generation of colored pathway maps KEGG data analysis	Open	[127]
**MetaboAnalyst**	Genomics Transcriptomics Proteomics Metabolomics	Plants, Microbial, Microbiome, Medical and Clinical	- Data processing and statistical analysis - Pathway analysis - Multi-omics integration	Open	[53]
**MetaboLights**	Metabolomics	Unspecified	- Database for storing metabolomics experiments and derived information - Database for cross-species and cross omics technique	Open	[106]
**MetScape 2**	Transcriptomics Metabolomics	Medical and Clinical	- Integrates data from KEGG and EHMN databases	Open	[128]
**mixOmics ** (R package)	Metagenomics Transcriptomics Proteomics Metabolomics	Unspecified	- Integration of data - Chemometric analysis (similarity/difference)	Open	[129]
**OmicsPLS**	Metagenomics Transcriptomics Proteomics Metabolomics	Unspecified	- Integration of data - Chemometric analysis (similarity/difference) - R-package with an open-source implementation of two-way orthogonal PLS	Open	[130]
**Omickriging ** (R package)	Transcriptomics Proteomics Metabolomics Fluxomics	Unspecified	- Integration and visualization of omics data	Open	[131]
**Omix visualization tool**	Transcriptomics Proteomics Metabolomics Fluxomics	Unspecified	- Integration and visualization of omics data	Annual license fee	[132]
**PaintOmics**	Transcriptomics Metabolomics	100 top species of different biological kingdoms	- Integration and visualization of transcriptomics and metabolomics data	Open	[133]
**PathVisio 3**	Transcriptomics Proteomics Metabolomics	Unspecified	- Integration of omics data - Visualize omics data based on common data nodes and interactions in the pathway	Open (Apache)	[134]
**ProMeTra**	Transcriptomics Metabolomics	Medical and Clinical	- Interactive visualizations of metabolite concentrations together with transcript measurements mapped on the pathways and GenomeMaps	Open	[135]
**Reactome**	Genomics Transcriptomics Proteomics Metabolomics	Unspecified	- Multi-omics data visualization - Metabolic map of known biological processes and pathways	Open	[110]
**Recon3D**	Genomics Proteomics Metabolomics	Human	- Computation resource and comprehensive human metabolic network model	Open	[80]
**SimCell**	Genomics Transcriptomics Proteomics Metabolomics	Unspecified	- Graphical modeling tool	Open	[56]
**SIMCA**	Metagenomics Transcriptomics Proteomics Metabolomics	Unspecified	- Integration of data - Chemometric analysis (similarity/difference)	Commercial	[136]
**VANTED** (Visualization and Analysis of Networks with related Experimental Data)	Metagenomics Transcriptomics Proteomics Metabolomics	Unspecified	- Comparison of multiple omics data sets - Visualization of metabolic maps - Correlation networks analysis	Open	[137]
**VitisNet**	Metagenomics Transcriptomics Proteomics Metabolomics	Grapes	- Integration of data - Visualization of connectivity	Open	[138]
**xCellerator**	Proteins Metabolomics	Unspecified	- Mathematica package designed to aid biological modeling	Open	[55]

**Table 3 metabolites-09-00076-t003:** List of databases that aid multi-omics data integration process.

Database	Omics	Domain	Functionality	Type of license	Reference
**ChEBI**	Metabolomics	Unspecified	- Metabolomics database and ontology	Open	[105]
***E. coli* metabolome database (ECMDB)**	Metabolomics	Microbial	- Annotated metabolomics and metabolite pathway database	Open	[97]
**FlyBase**	Genomics Transcriptomics	*Drosophila*	- Genes and RNA-seq data of different *Drosophila* spp	License	[95]
**GenBank (database)**	Proteomics	Numerous (over 100,000 organisms)	- Proteomics database - Open access, annotated collection of all publically available nucleotide sequences and their protein transitions.	Open	[103]
**Human Metabolome Database (HMDB)**	Metabolomics	Human	- Human metabolite and pathway database	Open	[102]
**KEGG**	Genomes Transcriptomics Proteomics Metabolomics	PlantsAnimalsMicrobes	- Collection of databases dealing with genomes, biological pathways, diseases, drugs, and chemical substances.	Open and licensed.	[109]
**Lipid Maps**	Lipidomics	Unspecified	- Lipids database, ontology, and resource for standards and protocols relating to lipidomics	Open	[107]
**Plant Metabolic Network (PMN)**	Genomics Proteomics Metabolomics	Plants	- Plant-specific database containing pathways, enzymes, reactions, and compounds	Open	[100]
**PRIDE**	Proteomics	Unspecified	- Proteomics database	Open	[108]
**ProMeTra**	Transcriptomics Metabolomics	Medical and Clinical	- Interactive visualizations of metabolite concentrations together with transcript measurements mapped on the pathways and GenomeMaps	Open	[135]
**Reactome**	Genomics Transcriptomics Proteomics Metabolomics	Unspecified	- Database for molecular details of signal transduction, transport, DNA replication, metabolism, and other cellular processes	Open	[110]
***Saccharomyces* genome database (SGD)**	Genomics	Microbe (yeast)	- Database for genome sequence of *Saccharomyces* spp	Open	[101]
**UniProt**	Proteomics	Unspecified	- Proteomics database	Open	[104]
**Wormbase**	Genomics	Helminths	- Database that contains information on biology, genetics, and genomics of *Caenorhabditis elegans* and other nematodes	Open	[96]
**Yeast metabolome data (YMDB)**	Metabolomics	Microbe (yeast)	- Metabolite database	Open	[99]

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
