# Peer review of "Systems Biology and Multi-Omics Integration: Viewpoints from the Metabolomics Research Community"

_metabolites, 2019, doi:10.3390/metabo9040076_

Round 1
Reviewer 1 Report
Overall remarks:
The article discusses a relevant topic, namely the challenges that experimentalists and modelers have to face in the integration of multi-omics biological data. More specifically, the review takes the perspective of a researcher who generates and analyzes metabolomics data, and aims to guide such a researcher through integration of multi-omics data. The paper is overall fluid to read and comprehensible and touches the most critical aspects of the subject. The structure of the paper could however be improved to make it clearer. The description of the methods, software tools and databases could be more clearly separated. Also, the paper appears inhomogeneous in its content, being much less complete on the aspects not strictly related to metabolomics, thus on genomics, transcriptomics and proteomic data production and analysis, and on computational modeling.
Section-specific remarks:
1. Introduction
• The terms ‘compatible’ and ‘compatibility’ are not well defined and their use can open to misinterpretations
2. Designing Experiments Suitable for Multi-omics Integration
• The “specific questions” presented at the beginning of the paragraph could be reformulated more synthetically and precisely. For instance, it is not clear what the authors mean with “boundaries”.
• Table 1: Inappropriate and confusing use of hyphen “-”
• Top-down and Bottom-up approaches. I would encourage the authors to use different terms to describe the two types of strategies presented, as it conflicts with their more frequent usage. To be clearer, traditionally when we speak of top-down strategies we mean a procedure by which one starts from the data (either genomic, transcriptomic, proteomic or metabolomic) to deduce some mechanistic information of the system.
3. Multi-omics Data Integration:
• “Systems modeling techniques are primarily interpretive or hypothesis testing tools intended to provide more detailed mechanistic insights”. Models do not necessarily provide mechanistic insights, often they are just a tool to describe mechanistic insight mathematically.
• I suggest to change the term “post-data analysis” to something more clearly interpretable
• Paragraph “Systems Modeling”: the overview of the main modeling approaches is largely incomplete. I suggest to provide a more comprehensive view.
• I would stress the fact that the COBRA toolbox can be used to create models, but is not “used to predict” by itself.
• “As seen from the above examples of systems modeling for multi-omics integration, almost all are based on some form of metabolic model or metabolic readout.” This information is not sufficiently supported in the paper.
• I find table 2 confusing: databases and analysis software are listed altogether in the same table
4. Challenges in Multi-Omics Integration:
o Regarding “The Nature of the Omics Datasets”. Could be more structured with respect to the order of the techniques the challenges refer to.
o The examples of a complete meta-data (like in plants and cell culture) would be better supported by some citations.
Author Response
We thank Reviewer#1 for all the constructive feedback on our opinion piece. We made it clear in the abstract and introduction that this is the summary of viewpoints from metabolomics researchers who attended a peer session on this topic at the recent ANZMET Metabolomics Conference in Auckland in September 2018. Here, we discussed in detail how different tools and software developed for metabolomics can aid the integration of multi-omics data. Therefore, details about other omics approaches were not discussed in depth. However, we would like to point out that important citations relating to these other omics techniques were provided in the manuscript. We have revised the text to make this explicitly clear. We believe that such a comprehensive discussion on each of these omics approaches may be a topic for another perspective/review article. We have carefully addressed all the comments provided by Reviewer #1 and made appropriate amendments as listed in the attached document.

Reviewer 2 Report
Pinu et al. present an informative and important discussion about the promises and challenges of multi-omics studies. To make some of the messages more clear, the authors should clarify a few statements.
line 64: "Similarly, ...": The authors should be more clear on what they are referring to when making the statement. Transcriptomics and proteomics are quantitative nowadays. For instance, in proteomics, there were varieties of quantitative strategies established during the last years such as TMT, SILAC, MaxLFQ... What is certainly true and important is to compare applicability and accuracy/ precision of quantification strategies (e.g. absolute vs relative quantification)
line 73: This statement partially contradicts the statements made in the paragraph before. The authors should give a better explanation of "why". One could even argue the other way around. Proteomics is more mature when it comes to automated data integration and interpretation (e.g. does not rely upon in-house libraries and manual curation of MS2 spectra matches, which more often than not is still considered the gold standard for metabolomics). The authors elaborate more on that topic around line 200. Perhaps they can clarify already here that metabolomics might -for many aspects of an omics study- provide a "save" guidelines for sample handling and processing.
line 128: While it is certainly important to consider biases from sample sources and processing the examples given are misleading. There is a variety of papers using FFPE for proteomics (see multiple papers published from Matthias Mann e.g.). The authors should revise that point with better examples.
line 156 and onwards: The authors could elaborate more on the relationship between variance in the data, statistical power, the degree of insights one can gain (e.g. the problem of overfitting when developing statistical models on data with large feature space and small numbers of replicates).
line212: typo: ", ,"
line 226: I know what the authors mean, yet they should rephrase. Linearity is not a requirement for correlation. They probably refer to the increasing number of possible relations which in turn increases the requirement for the number of replicates to explore these relationships.
line 370 onwards: The can be made for proteomics, too. It seems to be more a call for quantitative data instead of an example for the special role of metabolomics.
line 419: See my previous comments. This is certainly true but the authors should make clear which omes they refer to.
line 423 "error rates of <0.001%": considering the number of features (nucleotides) that error rate has to be applied to, the number of false positives might be pretty high. Moreover, the authors should clarify that many omics strategies e.g. proteomics are usually FDR controlled (e.g. protein and peptide identification fdr and adjusted p-values for statistical comparisons). One could even argue the DNA only provides a qualitative readout while the other omes actually provide quantitative information.
line 426 and onwards: The discussion about the differences is important. However, there is also a good reason why for instance proteomics and metabolomics cannot be directly compared in terms of the requirement for standards. In this example, the proteome is a chemically homogenous molecular landscape and there are quite robust targeted decoy strategies to identify peptides and proteins. For the metabolome that is much more difficult since the chemical heterogeneity is larger. For this and many other reasons, the requirements for example for standards cannot be compared without a more detailed discussion about the underlying chemistry and technology.
last paragraph: talking about the right incentive structure to nurture collaboration and leverage experience from scientist across different omes, the authors may want to consider to talk about the authorship policy. One major problem in biomedical science is the importance of the first and last authorship. This compromises the incentive to team up since each scientist's career relies on the accumulation of two important positions on papers.
Author Response
We thank Reviewer#2 for all the helpful feedback on our manuscript. We have taken into account all the comments and revised the manuscript as appropriate. Our replies to all the comments are provided in the attached document.
